. Pathogens

# Evaluating modes of influenza transmission (EMIT-2): Insights from lack of transmission in a controlled transmission trial with naturally infected donors

Jianyu Lai[1], Hamed Sobhani[2], Kristen K. Coleman[1], Shih-Han Sheldon Tai[1], Filbert Hong[1], Isabel Sierra Maldonado[1], Yi Esparza[1], Kathleen M. McPhaul[1], Shengwei Zhu[2], Don L. DeVoe[2,3], Justin R. Ortiz[4], Shuo Chen[5], Temima Yellin[6], Juan Manuel Carreno[6], Florian Krammer[6,7,8,9], Benjamin J. Cowling[10], Aubree Gordon[11], Wilbur H. Chen[4], Jelena Srebric[2], Donald K. Milton [1,3]*, for the EMIT-2 Study Team¶

1 Department of Global, Environmental, and Occupational Health, School of Public Health, University of Maryland, College Park, Maryland, United States of America, 2 Department of Mechanical Engineering, University of Maryland, College Park, Maryland, United States of America, 3 Fischell Institute for Biomedical Devices, University of Maryland, College Park, Maryland, United States of America, 4 Center for Vaccine Development and Global Health, University of Maryland School of Medicine, Baltimore, Maryland, United States of America, 5 Department of Epidemiology and Public Health, University of Maryland School of Medicine, Baltimore, Maryland, United States of America, 6 Department of Microbiology, Icahn School of Medicine at Mount Sinai, New York, New York, United States of America, 7 Center for Vaccine Research and Pandemic Preparedness, Icahn School of Medicine at Mount Sinai, New York, New York, United States of America, 8 Department of Pathology, Molecular and Cell-Based Medicine, Icahn School of Medicine at Mount Sinai, New York, New York, United States of America, 9 Ignaz Semmelweis Institute, Interuniversity Institute for Infection Research, Medical University of Vienna, Vienna, Austria, 10 School of Public Health, The University of Hong Kong, Hong Kong Special Administrative Region, China, 11 School of Public Health, University of Michigan, Ann Arbor, Michigan, United States of America

¶ A complete list of EMIT-2 Study Team members is provided in S1 Text.
* dmilton@umd.edu

## Abstract

A previous controlled human influenza transmission trial produced minimal transmission using nasal inoculation of an egg adapted virus. Therefore, we implemented a new trial with naturally infected Donors. We recruited healthy Recipients for four, two-week hotel quarantine cohorts and naturally infected, qRT-PCR confirmed Donors for two cohorts. Five Donors (mean age: 21; 80% female; two H1N1, three H3N2, one for cohort 24b and 4 for 24c, Jan-Feb 2024) exposed Recipients (mean age: 36; 54% female, eight in cohort 24b and 3 in 24c) in a hotel room with limited ventilation but a high air recirculation rate. We collected exhaled breath, ambient and personal bioaerosols, fomite swabs, and sera, and analyzed samples using dPCR and fluorescent focus assays, hemagglutination inhibition (HAI) assay, and enzyme-linked immunosorbent assay (ELISA). Compared with previously studied community-acquired influenza cases, we detected viral RNA (44%) and culturable virus (6%) less frequently and measured fewer viral RNA copies (79 – 8.9 × 10$^3$ copies/30-min) in

**Data availability statement:** The data that support the findings of this study are publicly available on the Open Science Framework repository: https://osf.io/z7uvm/.

**Funding:** This study was supported by NIAID Cooperative agreement U19 grant (5U19AI162130) to D.K.M., by the University of Maryland Baltimore, Institute for Clinical & Translational Research (ICTR) and the University of Maryland Strategic Partnership: MPowering the State (MPower) to K.K.C., and by gifts from The Flu Lab and Balvi Filanthropic Fund to D.K.M. The funders had no role in study design, data collection and analysis, decision to publish, or preparation of the manuscript.

**Competing interests:** The authors have declared that no competing interests exist.

Donors' exhaled fine aerosols. One of 23 surface swab samples was culture positive. At admission, 8 of 11 Recipients had HAI titers ≤10 but 9 of 11 had stronger binding antibody responses than Donors against vaccine strains corresponding to Donor viruses. No Recipient developed influenza-like illness, PCR-positive respiratory samples, or serological evidence of infection. Potential explanations and insights regarding lack of transmission include importance of cough and seasonal variation in viral aerosol shedding by Donors, of potential cross-reactive immunity in middle-aged Recipients with decades of exposure, and of exposure to concentrated exhaled breath plumes limited by rapid air mixing from environmental controls that distributed aerosols evenly. Future trials over multiple seasons, Donors that cough, younger recipients, and environments that preserve normal exhaled breath plumes will be required to observe transmission from naturally infected Donors under controlled conditions and generate new insights into influenza transmission dynamics.

## Author summary

Human-to-human influenza virus transmission under controlled conditions could provide insights leading to better control of epidemics and pandemics. However, a previous study using laboratory adapted viruses produced minimal transmission. Therefore, we aimed to study transmission from people naturally infected with circulating viruses. We recruited four cohorts of healthy volunteer Recipients to stay in a quarantine hotel for two weeks. We could not recruit Donors for the first two cohorts. In the last two cohorts, one Donor exposed eight Recipients in the first and four Donors exposed three Recipients in the second. The Donors coughed infrequently and shed less virus into the air than we had observed during previous influenza seasons. No Recipients became infected. Possible explanations include that people infected during mild influenza seasons or who cough very little may be minimally contagious. Our middle-aged Recipient cohorts were older than Donors and possibly less susceptible to infection because of additional years of vaccination and infection. Finally, environmental controls in the hotel distributed aerosols evenly but reduced short-range exposure to concentrated clouds of exhaled breath that may play an important role in transmission. New designs will need to address these issues.

## Introduction

In a typical year seasonal influenza is associated with hundreds of thousands of hospitalizations and tens of thousands of deaths in the US [1]. Although the 2009 pandemic influenza virus had a broadly similar severity profile to seasonal influenza [2–4], the three previous pandemics in the 20th century were associated with much greater morbidity and mortality [5]. Influenza is therefore responsible for a considerable burden of disease in the US and globally, and control of influenza is a public health priority. The effective reproductive number

of seasonal influenza is estimated to be around 1.3 at the start of an influenza season, meaning that we expect an average of 1.3 secondary cases from an infected person, depending on the environment and number of people whom they come into contact with [6].

It is now well established that humans shed infectious virus into fine particle aerosols both when coughing and also when simply breathing [7]. A post-hoc analysis of household studies suggests that viral aerosols play an important role in transmission, with half of influenza transmissions in households occurring via inhalation [8]. In April 2024, the World Health Organization released a technical consultation report and defined the modes of respiratory pathogen transmission as airborne transmission/inhalation, direct deposition, and contact (direct and indirect) corresponding to a previously proposed transfer-process categorization: inhalation, spray, and touch transmission, respectively [9,10]. Despite recent emphasis on airborne transmission [9,11], the relative importance of these transmission modes remains poorly understood and an important focus of continuing research interest [12,13]. The lack of in-depth understanding of transmission modes and mechanisms impedes development, implementation, and prioritization of effective control strategies.

In 2013, a randomized controlled trial (RCT) of human challenge-transmission, Evaluating Modes of Influenza Transmission (EMIT-1), was conducted to evaluate the relative importance of these transmission modes [14,15]. EMIT-1, using volunteers intranasally inoculated with $10^5$ 50% tissue culture infectious doses ($TCID_{50}$) influenza virus as Donors, failed to meet the expected secondary attack rate (SAR); only one serology-confirmed asymptomatic, PCR-negative infection was identified in a Control Recipient [15]. Post-hoc analyses revealed that Donors intranasally inoculated with influenza virus had mild symptoms and minimal shedding of virus in exhaled breath compared with symptomatic community-acquired cases on a college campus collected in the same year (2013) [14] as well as mildly symptomatic influenza cases from a longitudinal cohort [16]. These findings suggested that if lack of transmission was due to use of an egg-adapted challenge virus, a controlled human influenza virus transmission trial (CHIVITT) with community-acquired cases as Donors could be an improved and more valid approach to studying the dynamics of influenza transmission.

Here, we describe EMIT-2, a first attempt at a CHIVITT incorporating community-acquired cases, and the challenges, limitations, and the insights gained regarding the dynamics of influenza transmission.

## Results

A schematic of the study design is shown in Fig 1a and the timeline of two cohorts is shown in Fig 1b.

### Study cohort and study population

By January 3, 2024, we recruited 80 potential Recipients into a study registry (S1 Fig). A total of 505 potential Donors expressed interest, with six ultimately admitted to the quarantine facility (S2 Fig). Between 2023 and 2024, we ran four quarantine cohorts: one in 2023 (23a) and three in 2024 (24a, 24b, 24c). Recipients were recruited to participate in quarantine cohorts without further selection based on antibody levels or vaccination status, due to the small number available. Recipients were assigned to Intervention Recipient (IR) and Control Recipient (CR) groups using the face shield and hand hygiene intervention previously described for the EMIT-1 study that used nasally inoculated Donors [15].

Despite successful enrollment of Recipients in cohorts 23a and 24a (S1 Table), we were unable to recruit Donors with natural influenza virus infection. For Cohort 23a (February 20 to March 5, 2023), the start of which was delayed due to the coronavirus disease 2019 (COVID-19) pandemic, we could not recruit Donors because the influenza season was unusually early and had ended before the quarantine commenced. For Cohort 24a (January 3–16, 2024), one Donor recruited based on a positive rapid influenza test reported by an outside facility was later confirmed to be infected with a seasonal coronavirus (229E), not influenza virus (S2 Table).

The later two cohorts 24b and 24c included 11 Recipients and 5 naturally infected Donors. We enrolled three IRs, five CRs and one Donor in cohort 24b (Jan 24 to Feb 6, 2024). We enrolled one IR, two CRs, and four Donors in 24c (Feb 14–27, 2024). All Donors reported symptom onset within the last 48 hours. However, a rapid flu test result submitted after enrollment by one of the Donors (D24c-3), on closer examination, demonstrated that they had a positive test more than

**a**

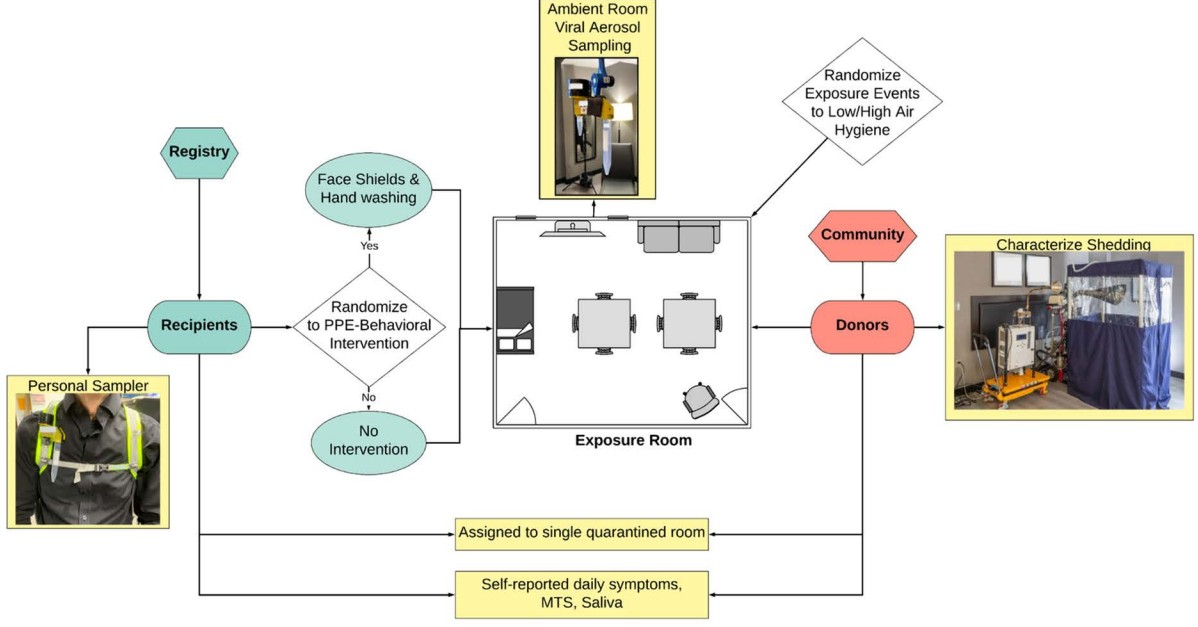

**b**

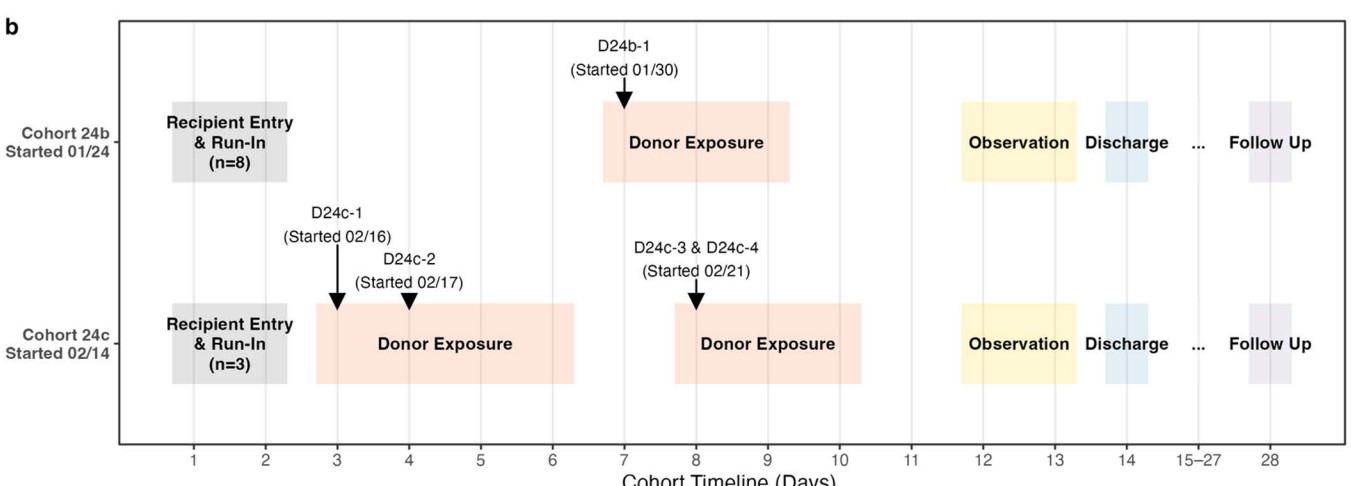

**Fig 1. Study design and timeline. (a)** Schematic of the study workflow and exposure event room. **(b)** Timelines for cohorts 24b (started January 24, 2024) and 24c (started February 14, 2024). Shaded blocks indicate phases (Run-in, Donor exposure, Observation, Discharge, Follow-up). Arrows mark donor introductions (D24b-1 on 01/30; D24c-1 on 02/16, D24c-2 on 02/17, D24c-3&D24c-4 on 02/21).

48 hours prior to enrollment. All tested positive for influenza A virus by a molecular test (Cepheid GeneXpert Xpress) at recruiting sites on the University of Maryland College Park campus or on arrival at the hotel quarantine. The Donor for 24b (D24b-1) and two of the Donors for 24c (D24c-1 and D24c-2) were infected with H3N2, while the other two Donors (D24c-3 and D24c-4) were infected with H1N1.

For cohorts 24b and 24c, 54% of the Recipients were female and the average age was 36 (range 22–49). Two Recipients (both in Cohort 24b) received an influenza vaccine earlier in the same influenza season before enrolling in a quarantine (Table 1). Among the five Donors, most were female (80%) young adults, with a mean age of 21 years (Table 2).

**Table 1. Recipients demographics for cohorts 24b & 24c.**

| | Cohort 24b | | Cohort 24c | | All |
|---|---|---|---|---|---|
| | CTL | INT | CTL | INT | |
| Number of participants | 5 | 3 | 2 | 1 | 11 |
| Female, N (%) | 2 (40) | 2 (67) | 1 (50) | 1 (100) | 6 (54) |
| Age, mean (SD) | 36.4 (10.5) | 29.0 (6.08) | 41.0 (4.24) | 39.0 (-) | 35.5 (8.56) |
| Vaccinated[a], N (%) | 1 (20) | 1 (33) | 0 (0) | 0 (0) | 2 (18) |
| Latino, N (%) | 0 (0) | 1 (33) | 0 (0) | 0 (0) | 1 (9) |
| Race | | | | | |
| Black or African American, N (%) | 2 (40) | 1 (33) | 1 (50) | 0 (0) | 4 (36) |
| White, N (%) | 3 (60) | 2 (67) | 1 (50) | 1 (100) | 7 (64) |

a. Vaccinated against influenza in the past 6 months at the time of entering the quarantine facility.

**Table 2. Donor demographics for cohorts 24b & 24c.**

| | Cohort 24b | Cohort 24c | All |
|---|---|---|---|
| Number of participants | 1 | 4 | 5 |
| Female, N (%) | 1 (100) | 3 (75) | 4 (80) |
| Age, mean (SD) | 23.0 (-) | 20.5 (0.577) | 21.0 (1.22) |
| Vaccinated[a], N (%) | 0 (0) | 2 (50) | 2 (40) |
| Latino, N (%) | 0 (0) | 2 (50) | 2 (40) |
| Race | | | |
| Asian, N (%) | 1 (100) | 0 (0) | 1 (20) |
| Black or African American, N (%) | 0 (0) | 2 (50) | 2 (40) |
| White, N (%) | 0 (0) | 2 (50) | 2 (40) |

a. Vaccinated against influenza in the past 6 months at the time of entering the quarantine facility.

## Exposure events and activities

We conducted 23 exposure events involving influenza Donors, including 6 events in cohort 24b and 17 events in cohort 24c. These events lasted between 111 and 250 minutes, totaling 82.2 cumulative influenza exposure-event hours (19.5 in 24b and 62.7 in 24c). Eleven events involved one Donor and 12 involved two (S3 Table). During the events, volunteers engaged in conversation-focused activities (such as icebreaker questions), interactive games (such as the card game UNO and drawing and guessing game Telestrations), plus some physically active periods (stretching, yoga, or dancing). A designated object was passed following a donor-centered alternating sequence, including a marker in 19, tablet computer in 2, and microphone in 2 events (see S3 Table).

## Environmental conditions

Mechanical systems successfully maintained stable environmental conditions in the exposure event room during exposure events through running two fan coil units and two large capacity dehumidifiers with fans constantly on high speed. The indoor air temperature remained within 22-25°C, and relative humidity ranged between 20–45% (Fig 2). The ventilation rate was approximately 0.25-0.5 air changes per hour. Running fan coil units and dehumidifiers at high speed in addition to portable air cleaners without filters served to create a well-mixed environment satisfying assumptions of the standard Wells-Riley model [17]. However, the extensive mixing reduced the extent of Recipient exposure to concentrated plumes of exhaled breath (a detailed computational fluid dynamic analysis will be published separately).

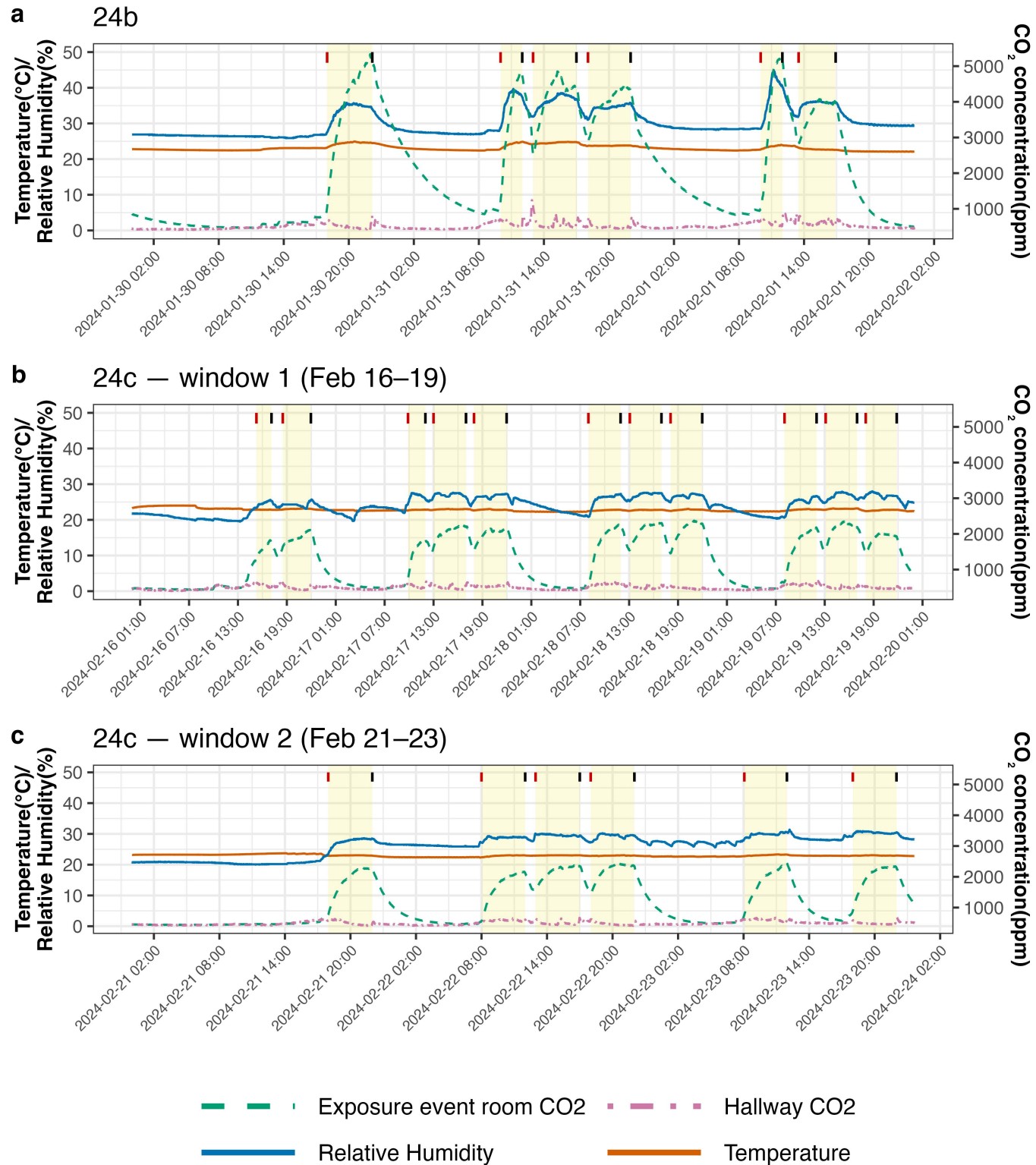

**Fig 2. Variations in indoor temperature, relative humidity, and $CO_2$ concentration in the exposure event room and hallway during cohorts. (a)** Cohort 24b (Jan 30–Feb 1, 2024); **(b)** Cohort 24c, first donor block (Feb 16–19, 2024); **(c)** Cohort 24c, second donor block (Feb 21–23, 2024). The left y-axis shows temperature (°C) and relative humidity (%); the right y-axis shows $CO_2$ concentration (ppm). Line plots show temperature (orange, solid),

relative humidity (dark blue, solid), exposure event room $CO_2$ (green, dashed), hallway $CO_2$ (purple, dot-dashed). Yellow bands denote each exposure event windows; vertical ticks mark exposure start (red) and end (black). Temperature and humidity were maintained within stable ranges by the mechanical systems, as these parameters can influence virus stability. $CO_2$ concentration is a proxy for ventilation and a direct indicator of rebreathed air; levels increased during exposure events due to occupancy and slowly returned to background afterward, reflecting the low ventilation condition targeted in these experiments.

$CO_2$ concentrations in the exposure event room increased rapidly during exposure events, as occupants were the sole $CO_2$ source, and returned toward the background level (~420 ppm) after the exposure event. In 24b, with 11 occupants (one Donor, eight Recipients, and two study moderators), $CO_2$ levels rose from 460 to 5300 ppm, while in 24c, with 7 occupants (two Donors, three Recipients, and two study moderators), $CO_2$ levels were lower, rising from 430 to 2400 ppm. Fluctuations in $CO_2$ correlated with variations in temperature and relative humidity. $CO_2$ levels in the corridor remained relatively stable near background concentrations, with minor fluctuations caused by occasional human traffic and the opening of the exposure event room doors.

### Viral shedding and environmental contamination by donors

**Exhaled breath aerosol samples.** Using the G-II, we collected 16 exhaled breath aerosol samples from five influenza Donors. During each 30-minute collection period, Donors coughed and sneezed spontaneously, with cough counts ranging from 0 to 16 (median: 0) and sneeze counts from 0 to 1 (median: 0) (S4 Table).

Among these exhaled breath aerosol samples, we detected viral RNA in seven (44%) fine (<5µm, range: 79 – 8.9 × 10³ copies per 30-minute sample) and one (6%) coarse (≥5µm, 53 copies per 30-minute sample) aerosol sample; all from Donors with H3N2 infections. The viral RNA load in breath aerosol samples fluctuated over time and did not follow a clear pattern across Donors (Figs 3 and 4).

The only culture-positive breath fine aerosol sample contained 147 focus forming units (FFU) per 30-minute sample (Figs 3 and 4). The sample came from the Donor in 24b (D24b-1; H3N2 infection) on the morning of Day 9 (February 1, 2024) their last day in quarantine, when they coughed the most during the 30-minute exhaled breath sampling (six times) and shed their highest breath aerosol viral RNA load (405 copies per 30-minute sample). Eight Recipients were exposed for four hours on this day.

**Mid-turbinate swabs and saliva.** We collected 18 mid-turbinate swabs (MTS) samples and 18 saliva samples from Donors. All MTS samples contained detectable viral RNA (range: $5.32 \times 10^4$ – $3.91 \times 10^8$ genome copies/swab), and 11 (61%) were culture positive with viral loads from 40 to $8.7 \times 10^4$ FFU/swab. Among saliva samples, 15 (83%) contained detectable viral RNA (range: 67 – $4.37 \times 10^7$ copies/mL); however, none were culture positive (Figs 3 and 4). We did not observe significant correlations between exhaled breath aerosol viral RNA and that measured in MTS or saliva samples in this small data set (S3 Fig).

**Ambient and personal bioaerosol sampling.** We collected two ambient air samples and two personal breathing zone air samples during each of 12 afternoon exposure events (one per day) when Donors were present, resulting in a total of 24 ambient and 24 personal samples.

Among the ambient samples, we detected viral RNA (75 genome copies/m³ air) by dPCR in one sample from cohort 24b, at sitting height (129 cm) in the 1–4 µm size fraction, on January 31, 2024 – the afternoon prior to the culture positive exhaled breath sample from Donor D24b-1. One other ambient sample, collected from cohort 24c (Donors D24c-1 & D24c-2 present; February 19, 2024) at standing breathing level height (150 cm), contained detectable viral RNA (25 RNA copies/m³ air) in the >4 µm size fraction (S4 Fig). None of the personal bioaerosol samples contained detectable viral RNA by dPCR.

**Surface swab viral load.** We collected 23 surface swabs from fomites used in exposure events with a Donor, including two from a microphone, two from a tablet computer, and 19 from a marker pen, depending on the activities

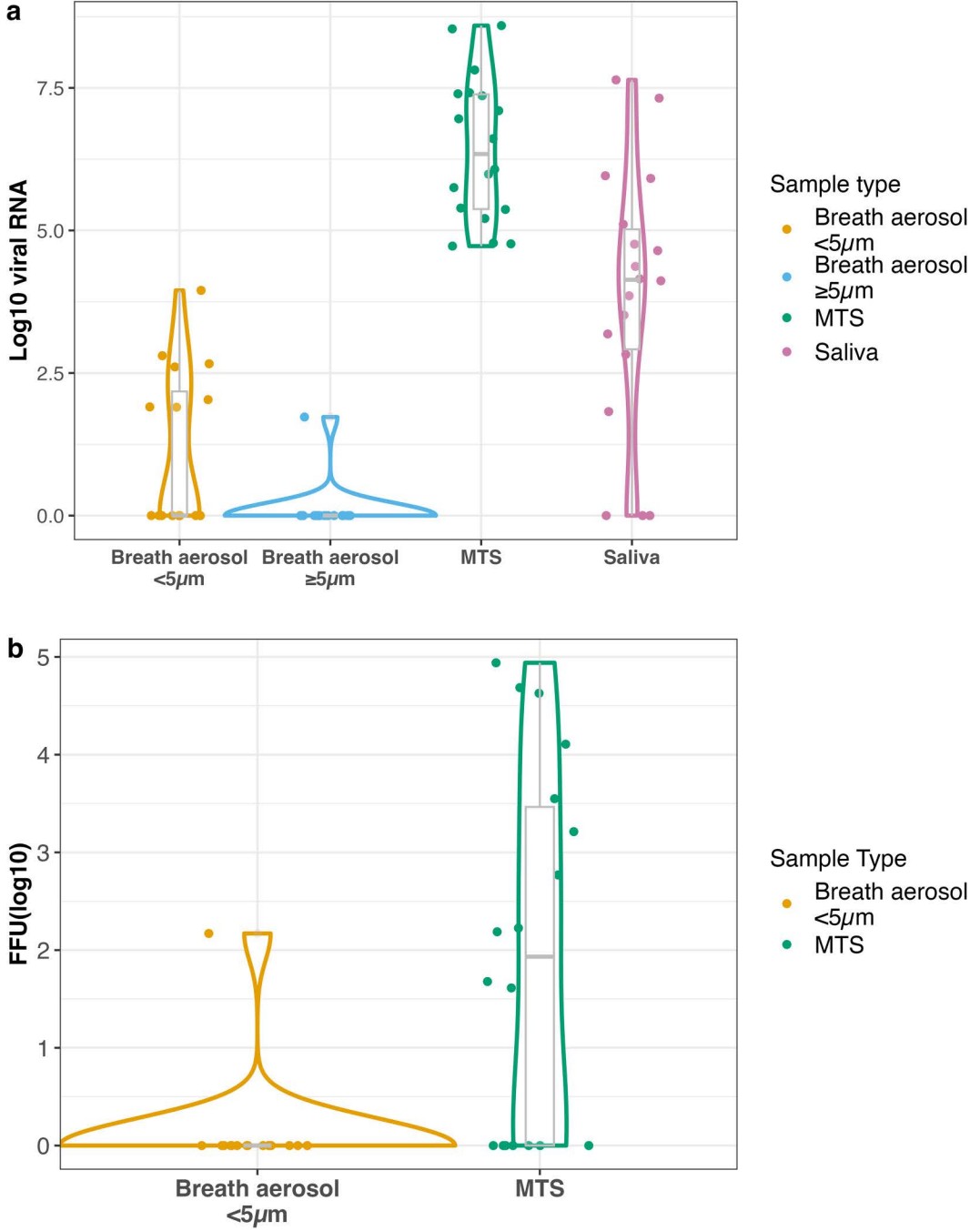

**Fig 3. Viral RNA and infectious viral load in Donor samples.** Violin plots depict the distribution of viral RNA and infectious (fluorescent focus unit, FFU) viral loads (log10 RNA copies and FFU) across four different sample types: breath aerosol <5 μm, breath aerosol ≥5 μm, mid-turbinate swabs (MTS), and saliva. RNA copy and FFU values are expressed as follows: per 30-minute sample for breath aerosols, per swab for MTS, and per mL for saliva. Negative samples were assigned a value of 1.

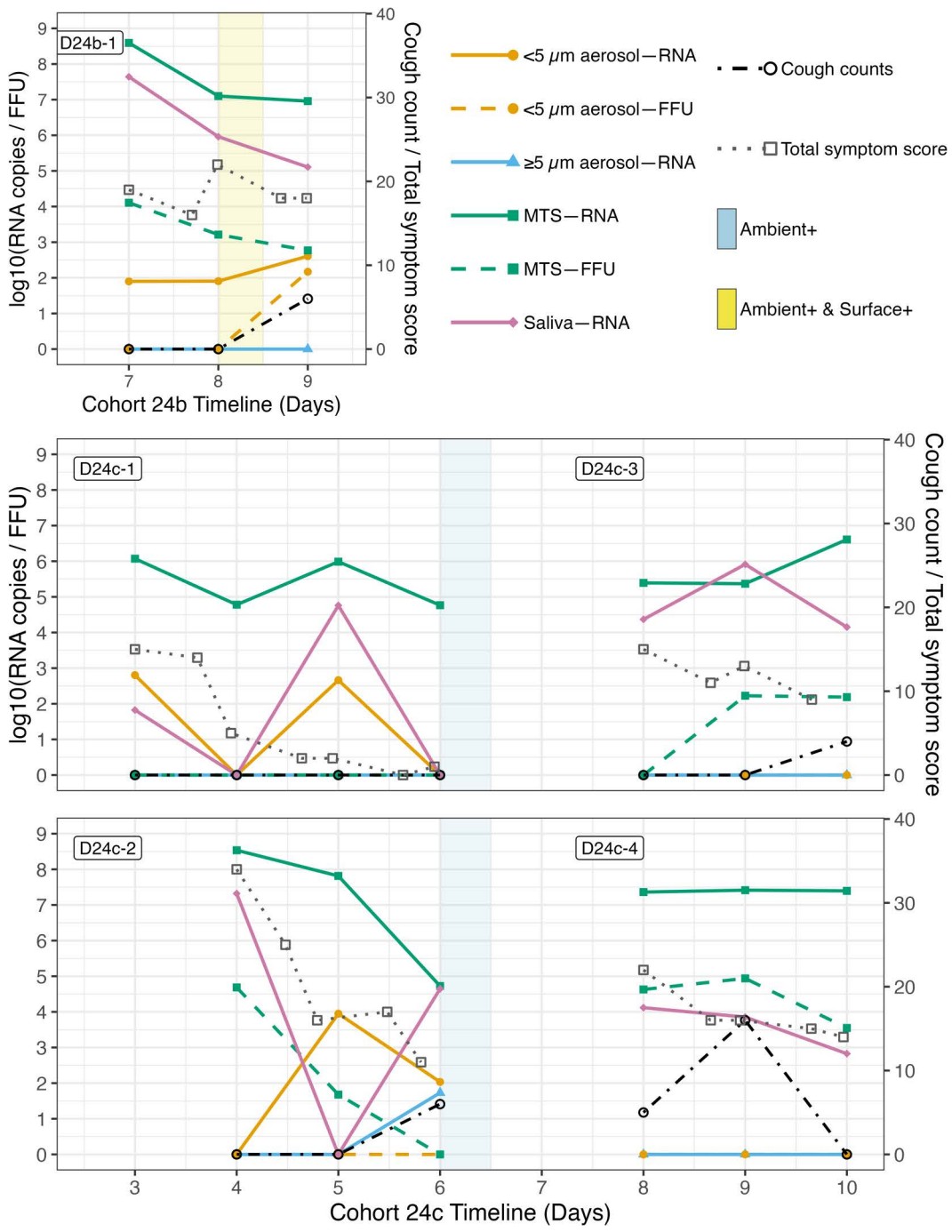

**Fig 4. Donor sample viral RNA, infectious viral load, total symptom score and cough trajectories with timing of positive environmental samples.** Each individual plot shows one donor over their days in quarantine. Lines/points show RNA (solid) and FFU (dashed) for: breath aerosol <5 μm (solid circle, orange), breath aerosol ≥5 μm (solid triangle, sky blue), mid-turbinate swab (MTS; solid square, green), and saliva (solid diamond, purple). Cough counts during 30-min breath sampling (dot-dash line, black, open circle) and self-reported total symptom score (dotted line, grey, open square) are overlaid and referenced to the right axis. Shaded bands mark environmental positives centered on the day that positives samples were detected: Ambient+ (light blue) and Ambient+ & Surface+ (yellow). Donors' samples were collected once per day, prior to any exposure events (on the afternoon or evening of admission and otherwise in the morning), ambient samples were collected during afternoon events and surface swabs at the end of each exposure event. RNA copy and FFU values are expressed as follows: per 30-minute sample for breath aerosols, per swab for MTS, and per mL for saliva. Negative samples were assigned a value of 1.

performed (S3 Table and S4 Fig). Among these, one sample, collected from a marker pen during the third exposure event of the Donor's second day in quarantine (24b), contained virus detected by dPCR; it was also positive for culture (total viral load of 454 RNA copies and 20 FFU).

### Symptoms reported by Donors and Recipients

Donors reported varying levels of symptom severity across a range of individual symptoms (Fig 5a). Throughout the quarantine period, total symptom scores fluctuated for the Donor in 24b (D24b-1), but generally declined for Donors in cohort 24c, though the rate of decline differed among Donors (Fig 5b).

Recipients sporadically reported individual symptoms, with overall symptom severity lower than that of Donors. Malaise was the most frequently reported symptom (Fig 6a). Total symptom scores fluctuated over time, with most Recipients reporting more symptoms after exposure to a Donor (Fig 6b). Three Recipients reported symptoms only on days with exposure events but not afterward. R24b-Ctl4 in cohort 24b recorded the highest total symptom score (10) three days after exposure to the Donor (Fig 6b).

### Recipient post-exposure PCR and culture

We collected a total of 89 post-exposure MTS and saliva samples from Recipients. In cohort 24b, each Recipient provided daily MTS and saliva samples on 7 days (quarantine cohort Days 8 through 14) following their first exposure event on Day 7. In cohort 24c, each provided 11 samples (Days 4 – 14) following their first exposure event on Day 3. None of the samples tested positive by dPCR.

### Serology from Donors and Recipients

Consistent with qRT-PCR test results, hemagglutination inhibition (HAI) titers for Donors measured at admission and follow-up confirmed three H3N2 (D24b-1 from cohort 24b; D24c-1 and D24c-2 from cohort 24c) and two H1N1 (D24c-3 and D24c-4 from cohort 24c) infections (Figs 7 and 8).

Among the exposed Recipients, 8 out of 11 had low HAI titers (≤10) at admission against the vaccine strains corresponding to the Donor strains, including 6 out of 8 in cohort 24b, 2 out of 3 in cohort 24c (Figs 7 and 8), and one of the two vaccinated for the current season. However, none of the Recipients showed a four-fold increase in HAI titers against the vaccine strains from admission to follow-up, except for one Recipient in cohort 24b (R24b-Ctl3), who responded to all vaccine strains and reported having received an influenza vaccine two days after discharge from quarantine.

The enzyme-linked immunosorbent assay (ELISA) results were generally consistent with the HAI titer findings, as none of the Recipients showed a four-fold increase in area under the curve (AUC) values against vaccine strains HAs corresponding to the Donor infections, including the Recipient who received an influenza vaccine after quarantine (S5 and S6 Figs). However, ELISA results indicated higher AUC values in Recipients than in Donors at baseline. In cohort 24b, all eight Recipients had higher AUC values at admission compared to the Donor. Similarly, in cohort 24c, one of three Recipients had a much higher AUC value than their Donor counterparts at admission.

## Discussion

We developed a novel framework for investigating human influenza virus transmission using Donors with community-acquired infections under controlled conditions. However, despite enrolling five naturally infected Donors within 48 hours of reported symptom onset, promoting prolonged close contact under low ventilation conditions with 11 Recipients, many with low HAI titers, no secondary infections were observed. Three possible explanations related to Donor source strength, Recipient susceptibility, and rapid air mixing may provide insight into influenza transmission dynamics and guide future study designs.

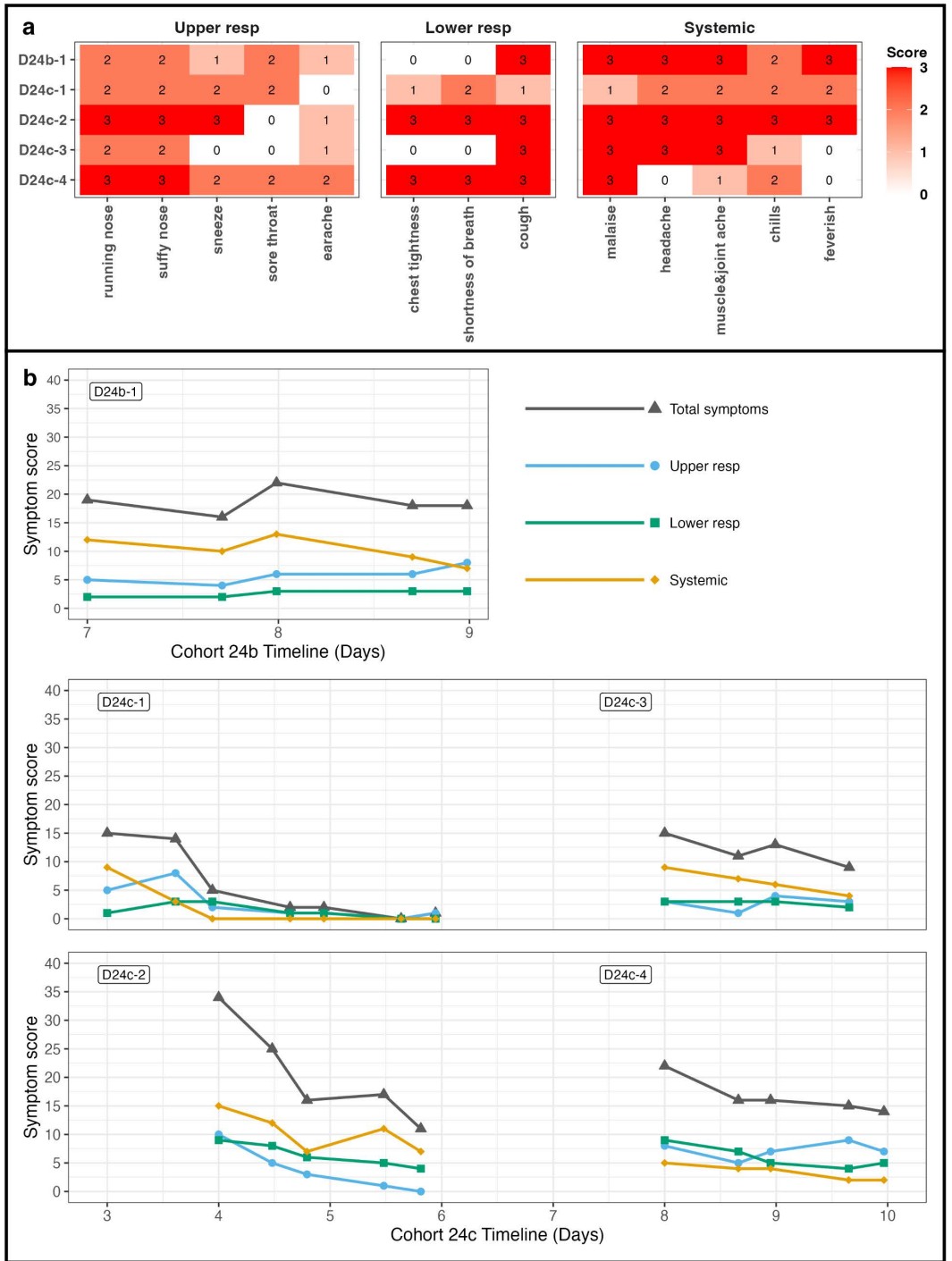

**Fig 5. Symptom severity and progression in Donors during quarantine. (a)** Heatmap shows the maximum individual symptom scores reported by individual Donors. The color intensity represents the symptom severity, with scores ranging from 0 to 3. Symptoms are listed along the x-axis, and Donors are on the y-axis. Columns are grouped by domains: Upper respiratory (running nose, stuffy nose, sneeze, sore throat, earache), Lower respiratory (chest tightness, shortness of breath, cough), and Systemic (malaise, headache, muscle & joint ache, chills, feverish). **(b)** Line plots present total symptom scores and symptom group scores (upper respiratory, lower respiratory, and systematic symptoms) over time for each individual Donor. The x-axis represents cohort timelines, and the y-axis indicates the sum of each symptom group scores.

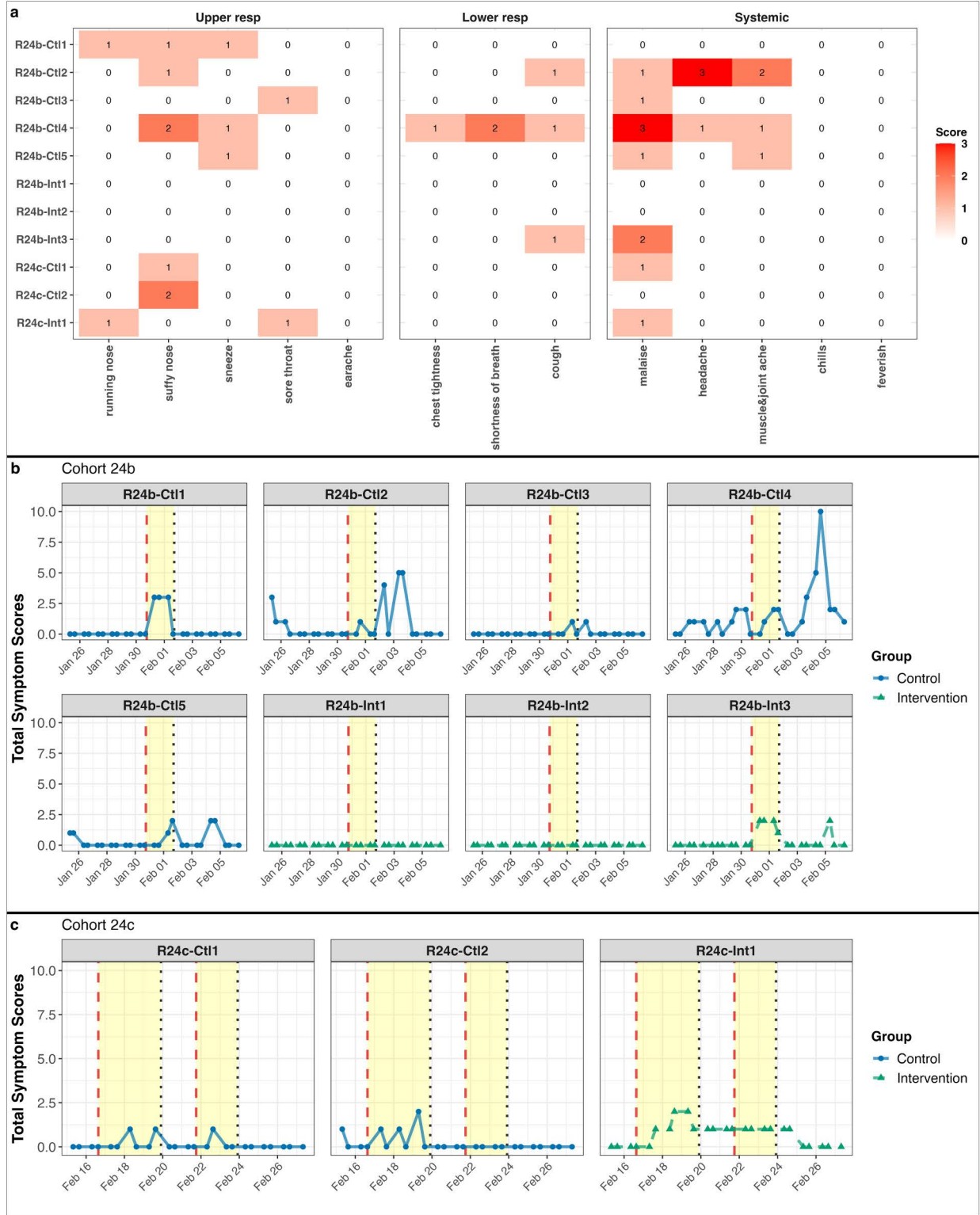

**Fig 6. Symptom severity and progression in Recipients during quarantine. (a)** Heatmap shows the maximum individual symptom scores reported by individual Recipients. The color intensity represents the symptom severity, with scores ranging from 0 to 3. Symptoms are listed along the x-axis, and Recipients are on the y-axis. **(b, c)** Line plots present total symptom scores over time for Recipients. (b) shows data for the Recipients in cohort 24b,

while (c) shows data for those in cohort 24c. The x-axis represents the timeline, with the yellow-shaded areas indicating periods of Donor quarantine. Each subplot corresponds to an individual Recipient, with blue lines (circle) representing the control group and green lines (triangle) representing the intervention group.

Donors in our present study, EMIT-2, shed detectable viral RNA somewhat more frequently, but not significantly more frequently than had been observed in EMIT-1 Donors infected with an egg-adapted laboratory virus: 44% (7/16) versus 22% (14/64) of fine and 6% (1/16) versus 9% (6/64) of coarse aerosol samples in EMIT-2 and EMIT-1, respectively [16]. EMIT-2 Donor aerosol shedding frequency was on the low end of previous observations of natural influenza infections, which ranged from 38% to 86% for fine aerosols and 29% to 86% for coarse aerosols [7,16,18]. The rate of viral RNA shedding into fine aerosols (≤5 μm) by EMIT-2 Donors was also similar to rate from the small fraction of experimentally infected Donors in EMIT-1 who had detectable aerosol shedding (range $1.6 \times 10^2$ to $1.8 \times 10^4$ copies per hour in EMIT-2 v. $2.6 \times 10^2$ to $1.6 \times 10^5$ copies per hour in EMIT-1) [15,19].

Looking specifically at symptomatic ILI cases on our campus infected with H3N2 from the 2012–13 and 2018–19 seasons, we previously noted that cases recruited during milder influenza seasons may, on average, shed lower quantities of viral RNA into exhaled breath [16]. We also noted that cases recruited by intensive case finding had milder symptoms and shed less frequently into aerosols. The results here are consistent with those earlier observations in that 2023–24 was a relatively mild season, most Donors were recruited by offering free tests at a kiosk on campus, and Donor D24b-1, with the strongest evidence for potential infectiousness, was the only donor recruited from the University Health Center.

Based on evidence for transmission to one Recipient in EMIT-1, Bueno de Mesquita et al. estimated a lower bound for an infectious quantum (i.e., the ID 63% via inhalation [20]) of approximately $1.4 \times 10^5$ RNA copies [19]. Recent aerosol inoculation experiments suggest that, with an egg-adapted laboratory strain, the infectious dose in approximately $10^4$ $TCID_{50}$. During EMIT-2, using an average minute-ventilation rate of 16 L/min based on Recipient activities and the two positive NIOSH bioaerosol samples, we estimate that on January 31, Recipients in Cohort 24b were exposed to approximately 750 RNA copies each. On February 19, each Recipient in Cohort 24c received an exposure of approximately 290 RNA copies (S4 Text). These exposures fell well below the infectious dose estimated from EMIT-1 and recent aerosol inoculation experiments. Therefore, low overall contagiousness of our Donors may be a key reason why no transmission was observed in this study.

This occurred despite two Donors (D24b-1 and D24c-2) having low Ct values from mid-turbinate swabs tested by Cepheid at admission to the hotel quarantine: 12.1 and 16.1 for D24b-1, who was recruited from the University Health Center, and 12.2 and 14.6 for D24c-2. We previously reported that viral load in the nose did not reliably predict exhaled virus concentrations [7]. Donor D24b-1 did not report cough and only coughed during the last of three exhaled breath collections – when we detected infectious virus in their breath. Similarly, Donor D24C-2 only coughed during the last exhaled breath collection. Cough observed during sample collection is a major predictor of viral shedding into aerosols. For future controlled studies of transmission from natural infections, it may be necessary to focus on recruiting Donors who cough.

In addition, we previously observed substantial heterogeneity in viral aerosol shedding in studies of community-acquired influenza and SARS-CoV-2 infections [7,16,21]. Recent human challenge studies have reported similar variability. For example, in a SARS-CoV-2 challenge, two participants accounted for 86% of airborne virus released, despite many being infected [22]. Similarly, a recent influenza challenge study found wide variation in the aerosol emissions of infected donors [23]. These consistent findings across both naturally acquired and controlled infections underscore the importance of inter-individual variability in aerosol shedding. Such heterogeneity likely contributes to the overdispersion of transmission seen in real-world outbreaks and may help explain the generally low aerosol shedding observed in our Donors.

The absence of transmission may also be attributable to Recipient pre-existing immunity. Although most of the Recipients had low HAI titers (≤10) against the vaccine strains corresponding to Donor viruses, their ELISA AUC values against these strains were higher at baseline than those of Donors, suggesting potential for some degree of pre-existing immunity

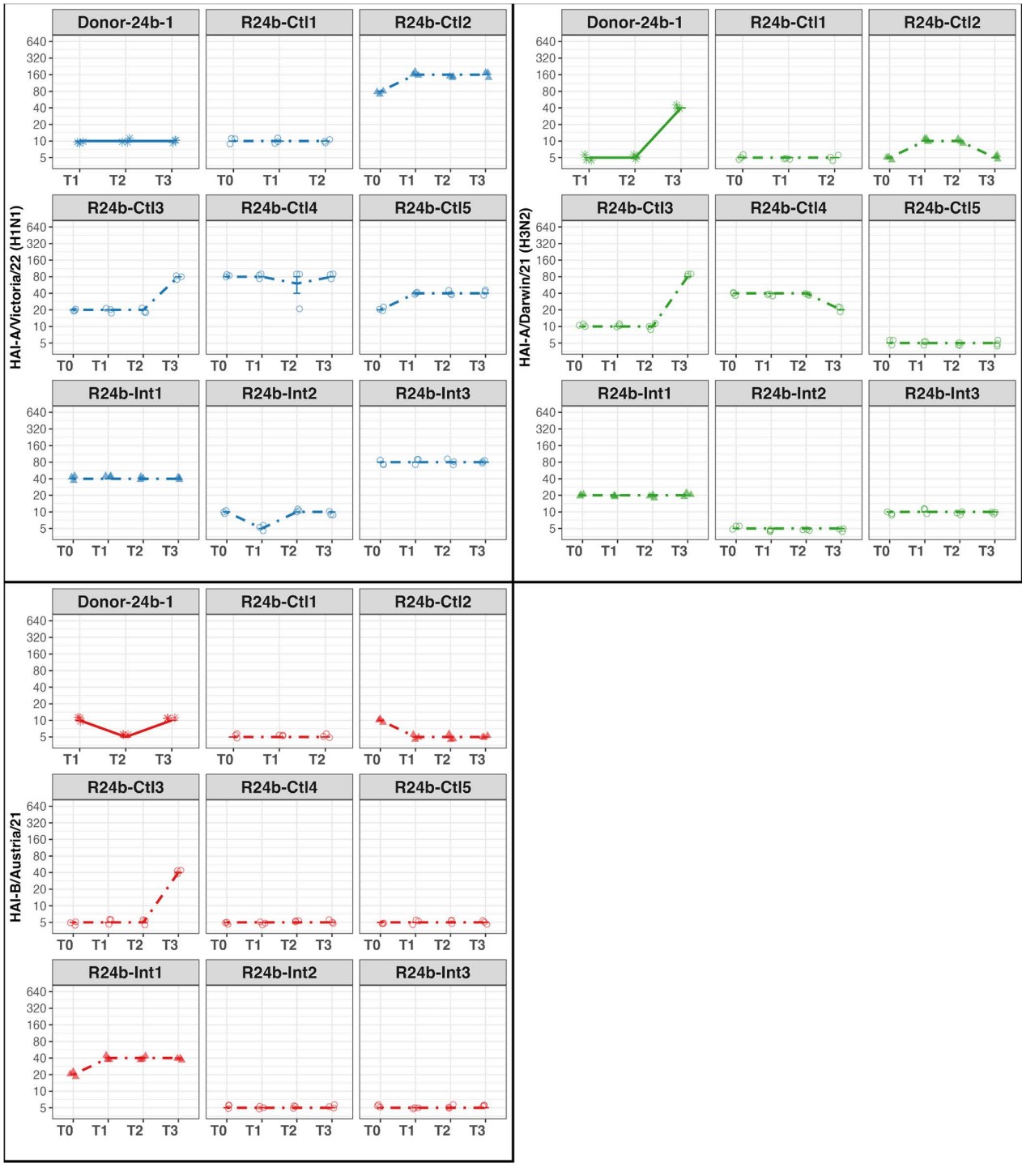

**Fig 7. HAI titers over time for Donors and Recipients in Cohort 24b.** HAI titers are shown for Donors (solid lines, star symbols), Vaccinated Recipients (dot-dashed lines, filled triangles), and Unvaccinated Recipients (dot-dashed lines, open circles) at four time points: T0 (Recipient screening), T1 (Admission), T2 (Discharge), and T3 (Follow-up). Virus targets are color-coded: A/Victoria/4897/22 (H1N1, blue), A/Darwin/6/21 (H3N2, green), and B/Austria/1359417/21 (red). Each facet represents an individual study participant.

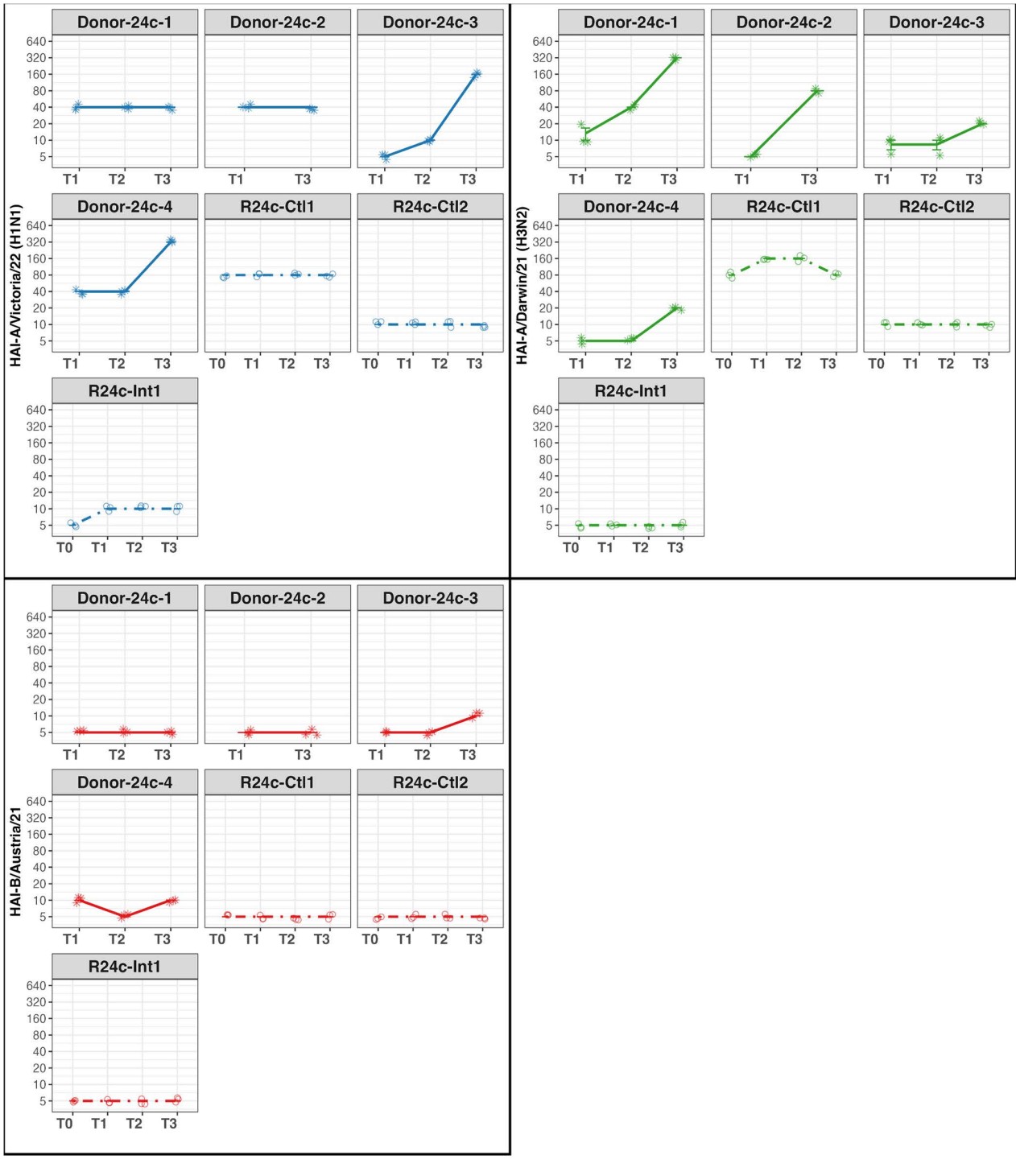

**Fig 8. HAI titers over time for Donors and Recipients in Cohort 24c.** HAI titers are shown for Donors (solid lines, star symbols) and Unvaccinated Recipients (dot-dashed lines, open circles) at four time points: T0 (Recipient screening), T1 (Admission), T2 (Discharge), and T3 (Follow-up). Virus targets are color-coded: A/Victoria/4897/22 (H1N1, blue), A/Darwin/6/21 (H3N2, green), and B/Austria/1359417/21 (red). Each facet represents an individual study participant.

from prior infections or vaccinations. The EMIT-1 challenge study began with a registry of 496 volunteers with HAI titers ≤10 against the challenge virus, who reported not having had a seasonal influenza vaccine in the last 3 years [15]. In contrast, our current study had a much smaller registry pool (80 volunteers). Over one-fourth of Recipients whom we later exposed to an influenza Donor had baseline HAI titers >10 and nearly one-third of registry members received an influenza vaccine during the same influenza season as the quarantine, including two who were later exposed to an influenza Donor. Given that our Recipients were middle-aged adults, they would have had greater cumulative exposure to influenza virus infections and vaccinations than our Donors, further contributing to broader immunity. Older adults are more likely than children to develop enhanced pre-existing cross-reactive antibodies due to prior infections and vaccinations [24], and middle-aged adults are less likely than young adults to show evidence of infection in large population studies [25]. Therefore, future transmission trials should focus on recruiting younger adults into the pool of volunteer recipients.

Finally, temperature and humidity during exposure events were within the range likely to promote transmission and therefore do not explain the lack of observed transmission in this study. Laboratory and animal studies reviewed by Arundel et al. [26] suggest that influenza virus survival and transmission are enhanced at low relative humidity (<40%) and may increase again at high humidity (>70%), with a mid-range of ~40–70% minimizing combined infectivity and survival. According to standards from the American Society of Heating, Refrigerating and Air-Conditioning Engineers (ASHRAE), thermal comfort for most individuals lies between 67–82 °F (19.4–27.8°C) [27]. In our study, temperatures were maintained at 22–25°C and relative humidity ranged from 20% to 45%, a range that supports participant comfort while enabling influenza survival and transmission.

Although the relationship between ventilation and airborne transmission is complex, low air change rates have been linked to increased risk in healthcare and public settings [28]. Menzies et al. [29], for example, observed higher tuberculosis infection risk in hospital rooms ventilated at <2 air changes per hour (ACH). While no universal minimum threshold has been validated, we maintained ventilation at ~0.25–0.5 ACH to support transmission. To achieve temperature and relative humidity control in the hotel room used for exposure events, in the presence of low ventilation and seven or more occupants, we ran the room's two fan coil units at maximum speed and added two large capacity portable dehumidifiers. Operation of these fans produced high rates of air mixing.

Controlled test chamber and modeling studies show that high rates of air mixing reduce short-range aerosol exposure [30,31] by rapidly breaking up concentrated plumes of exhaled breath. Exposure to concentrated plumes may be a critical factor in transmission via aerosol inhalation, especially from hosts with low to moderate viral aerosol shedding rates. Given the low viral aerosol shedding rates of our Donors, rapid air mixing with reduced short-range aerosol exposure may have contributed to lack of transmission in this study. Future studies using an indoor climate facility designed for indoor air quality and climate studies and capable of controlling temperature and humidity with controlled, minimal air movement and ventilation could avoid this problem.

Together, the three possible explanations for lack of transmission in this study (source strength, susceptibility, and air mixing) may provide insight into the transmission of seasonal influenza. For seasonal influenza, previous studies suggest that 4–7 out of 10 cases shed some virus into aerosols. But the geometric mean shedding rate per hour observed was at most one tenth of the infectious dose estimated from EMIT-1 for a selected population with minimal immunity. Although the average infectious dose for a population with some cross-reactive immunity may be higher, susceptibility to circulating human virus without laboratory adaptation is likely lower. Therefore, an infectious dose associated with exhaled breath aerosols in the rage of $10^4$ to $10^5$ viral RNA copies seems reasonable. Shedding rates are lognormally distributed and the upper tail extended to $10^7$ to $10^8$ RNA copies/hour, suggesting that some cases rare cases may shed as many as 100–1000 infectious quanta per hour [7]. However, reported geometric mean aerosol shedding rates suggest that only more symptomatic cases with cough would be expected to shed one or more infectious doses per hour and that mild cases would rarely be contagious [16]. This evidence for influenza virus supershedding suggests that while the average symptomatic case may be able to transmit to someone through prolonged exposure to concentrated breath plumes, a few

cases may be able to transmit relatively rapidly with less intense and prolonged exposures. An additional implication may be that prolonged face-to-face contact and a moderate degree of superspreading combine to generate seasonal influenza epidemics.

These considerations suggest that future controlled experiments aiming to observe transmission from naturally infected cases focus on recruiting Donors with cough and young adult volunteers with low rates of prior immunity as Recipients. It also points to advantages of a controlled environment with limited air mixing that allows exhaled plumes to linger, such as certain displacement ventilation arrangements [32].

The unpredictable timing and duration of influenza seasons pose a challenge for conducting a CHIVITT using Donors with community-acquired infection. The inability to recruit an influenza virus-positive Donor in 2023, due to an early and short influenza season, highlighted the need to prepare for potential quarantines from November through early March to capture both early and late influenza seasons, as we did for the 2023–24 influenza season. The resulting prolonged hotel rental including months without influenza activity, added considerable expense. New more flexible designs that can respond to timing of seasonal influenza activity without tying up resources for prolonged periods are needed. An approach that can leverage existing campus-based indoor environment research infrastructure and also study other respiratory viruses when influenza is not actively circulating could be cost effective. Another limitation was that Control Recipients may have reduced behaviors such as touching their face or eyes, which are relevant to touch-contact transmission. To better ensure potential for occurrence of touch-contact transmission additional behavioral requirements, such as instructing Control Recipients to touch their face at regular intervals as previously described for rhinovirus studies [33], could be incorporated and create a clear contrast with the hand hygiene intervention. Finally, we note that mucosal immunity, both innate and acquired, play an important role in susceptibility to initial infection and were not examined in this study.

## Conclusions

Lack of transmission from mild influenza cases, who rarely coughed, to healthy middle-aged volunteers in a room with rapid air mixing may provide some insights into influenza transmission dynamics. Transmission from healthy young adults may be dominated by the symptomatic cases with cough. Consistent with population data, middle-aged adults may be relatively immune to circulating influenza viruses. Finally, concentrated breath plumes may play an important role in transmission of seasonal influenza viruses. Future studies addressing these three factors are needed to advance our understanding of the role of inhalation exposure in transmission of seasonal influenza viruses.

## Materials and methods

### Ethics statement

The study was registered at ClinicalTrials.gov (NCT05666245) and approved by the Institutional Review Board of the University of Maryland, Baltimore (IRB protocol #HP103865). All participants provided written informed consent prior to enrollment, with assurances of confidentiality and the right to withdraw at any time without consequence.

### Overview of study design

In this CHIVITT our study population consisted of two groups of volunteers – healthy volunteer Recipients without evidence of respiratory infection on a multiplex molecular test, and Donors with recent-onset community-acquired influenza virus infection. Recipients consented to stay in a quarantine hotel for two weeks, while Donors agreed to stay for two to five days. Donors and Recipients interacted during exposure events in an exposure-event room sealed to provide low ventilation. The Recipients were randomly assigned to intervention Recipient (IR) or control Recipient (CR) groups. IRs wore a face shield to interrupt direct deposition transmission and sanitized their hands every 15 minutes to interrupt direct and indirect contact (touch) transmission; IR and CR groups in each exposure event received similar risk from inhalation

of infectious respiratory particle aerosols. CRs did not wear face shield or sanitize their hands during the exposure event; they would have been exposed to all modes of transmission (Fig 1).

### Study setting

This study took place in a dedicated, isolated floor of a hotel in Baltimore, Maryland, USA that had been used as a quarantine hotel for COVID-19. We equipped 18 rooms for individually housing up to 16 Recipients and 2 Donors.

### Volunteer recruitment

We recruited healthy volunteers aged 18 through 49 to enroll in a Recipient registry. Registry members who remained eligible, available, and willing to commit to a two-week stay in the hotel quarantine environment were admitted as Recipients at the start of each cohort study.

Concurrently, community members aged 18 through 59 with PCR-confirmed influenza virus infection and symptom onset within 48 hours were recruited and admitted to the quarantine facility as Donors. Eligibility criteria of Donors and Recipients are detailed in the Supporting Information (S3 Text).

### Randomization, concealment, and blinding

Recipients were randomly assigned to either the intervention group, receiving a hand hygiene and face shield intervention (HH + FS) as an IR, or the control group, receiving no intervention as a CR. We used randomization tables generated by a statistician, which were uploaded into the randomization module in REDCap (Research Electronic Data Capture) [34] to assign Recipients. REDCap was also used for data collection and management for this study.

Recipients and staff were unaware of the intervention assignment at enrollment; however, concealment during the study was not possible due to the nature of the intervention. All research nurses who collected biological samples and laboratory technicians who conducted PCR and serologic assays were blinded to the Recipients' assignments.

### Exposure event

During each exposure event, Donors and Recipients participated in structured activities intended to elicit sustained verbal interaction and standardized touch contact with a designated object passed from Donors to Recipients. Activities ranged from physically demanding exercises such as yoga and dancing to more sedentary activities like solving puzzles and watching movies. Each exposure event included activities that required participants to handle a shared object (marker, tablet computer, or microphone), during which a donor-centered alternating sequence was used: one Donor handed the object to a Recipient, the object was returned to a Donor, and the Donor then passed it to the next Recipient continuing until all Recipients had participated before beginning a new round. Two staff members were present at the exposure event room to observe, document, and moderate activities.

### Symptom monitoring

Both Donors and Recipients assessed their symptom twice daily by answering a survey, Modified Jackson Score, in their mobile device. They self-reported on a scale of 0–3 (0 = "no symptom", 1 = "Just noticeable", 2 = "Bothersome from time to time, but doesn't prevent me from doing activities", and 3 = "Bothersome most or all the time, and prevents me from doing activities") for the following symptoms: running nose, stuffy nose, sneezing, sore throat, earache, malaise (tiredness, fatigue), headache, muscle and/or joint ache, chills, feverish, chest tightness, shortness of breath, and cough.

## Biological sample collection

Mid-turbinate swabs (MTS) and saliva samples were collected daily from both Donors and Recipients. Blood samples were collected from both groups at three key time points: admission (at the start of the quarantine cohort, before any exposure event), discharge (at the end of the quarantine cohort), and a 28-day follow-up.

## Exhaled breath sampling

Donors provided exhaled breath samples using a Gesundheit-II (G-II) exhaled breath aerosol sampler [35] on admission to the quarantine hotel and each morning thereafter. G-II samples were collected for a 30-minute period with the volunteers breathing normally, while shouting and singing loudly three times at 5, 15, 25 minutes, as previous described [21]. The G-II samples were categorized into two aerosol size fractions, including fine (≤5 μm in diameter) and coarse (>5 μm in diameter) aerosols.

## Ambient and personal bioaerosol sampling

To collect ambient air samples during exposure events, we used a ceiling track system to position bioaerosol sampling devices developed by the National Institute for Occupational Safety and Health (NIOSH BC-251) [36] at sitting and standing breathing level heights (129 cm and 150 cm) in the designated exposure event room (Fig 1). To measure personal, breathing zone influenza virus aerosol exposure, we mounted NIOSH BC-251 samplers to the front strap of small backpacks containing a compact SKC AirChek TOUCH pump (Fig 1a). Each sample pump was calibrated to 3.5 L/min and all samples were collected for 4 hours (840 liters). We sampled one 4-hour exposure event per day for both ambient and personal influenza bioaerosols. For each sampled exposure event, we randomly selected one CR and one IR to wear a sampling kit. We extracted bioaerosol samples in 1 mL of extraction buffer and quantified viral RNA. To calculate RNA concentration per cubic meter of air, we divided the viral RNA copy number in each 1-mL sample by the total sampled volume (0.84 m$^3$), yielding RNA copies per m$^3$ of air.

## Surface swab sampling

During each exposure event, a selected object (marker, microphone, or a tablet computer, 10.2-inch iPad) used in planned activities was passed between Donors and Recipients. At the end of the exposure event, the object was swabbed for surface sampling before decontamination.

## Environmental interventions and infection prevention

The interactions between Donors and Recipients took place in a specially prepared exposure-event room in the quarantine hotel. To control temperature and humidity during high room occupancy, we operated the room's two built-in recirculating fan coil units equipped with MERV-8 filters on continuous high fan mode and installed two dehumidifiers. We also sealed major uncontrolled air pathways, including windows, doors and a leak in the fan coil units, prior to the quarantine cohorts to establish low ventilation, poor air hygiene conditions. To ensure that aerosols were distributed evenly and that mixing conditions did not change for exposure events when we used portable air filters to create high air hygiene conditions in future experiments, we ran the portable filter units without filters at the planned low-moderate fan speed.

We continuously monitored $CO_2$ concentration, indoor air temperature, and relative humidity in the exposure event room during each exposure event using eight uniformly distributed Aranet4 pro sensors, positioned on the walls at 1.0 and 1.5 meters above the floor, and two sensors suspended from the ceiling at a height of 2.1 meters. Additionally, we installed two Aranet4 pro sensors in the hallway near the exposure event room doors at a height of 1.5 meters. Using $CO_2$ concentration results, we calculated the ventilation rate of the room based on the concentration decay method [37].

To mitigate the risk of airborne inhalation/transmission outside of the exposure event room, we converted the entire floor into a quarantine facility through implementing an effective air cleaning strategy, incorporating 222-nm GUV fixtures and portable HEPA air cleaners in the hallways, and in-room portable HEPA air cleaners in the rooms other than the exposure event room (details described previously [38]). We also covered all the bathroom exhausts on the floor with MERV-16 filters and installed a portable HEPA/carbon filter unit in the bathroom of the exposure event room. Both Donors and Recipients stayed in assigned individual rooms equipped with in-room portable HEPA air cleaner and air curtains over the doorways when not participating in an exposure event. The room used for blood and swab collection was equipped with a portable HEPA filter, 222-nm GUV, and a local HEPA filtered exhaust for the participant's breathing zone. Donors and Recipients wore duckbill N95 respirators when outside of their rooms except during exposure events.

## Laboratory analysis

**BioFire and Cepheid GeneXpert Xpress tests.** Recipient admission and day 2 samples were submitted to the University of Maryland Hospital clinical laboratory for BioFire respiratory panel testing. Donor admission and daily Recipient samples, starting after exposure to Donors, were tested at the hotel quarantine using a Cepheid GeneXpert Xpress Cov-2/Flu/RSV plus 4-plex test panel.

**Digital PCR (dPCR) and fluorescent focus assay.** Total nucleic acids were extracted from G-II, mid-turbinate swabs, saliva, surface swabs, and ambient air samples for viral RNA detection. Influenza A virus RNA was quantified using recently updated primers and probes targeting the matrix (M) gene segment of influenza A viruses [39] and the QIAcuity One dPCR System (Qiagen). To assess viral infectivity, we measured focus-forming units (FFU) in G-II fine aerosol samples, mid-turbinate swabs, and surface swabs using humanized Madin-Darby canine kidney cells [40], adapting a previously described method [7]. Further details on sample processing and nucleic acid extraction are provided in Supporting Information (S4 Text).

**Serology.** Blood samples were analyzed at the Icahn School of Medicine at Mount Sinai, NY, USA. Serum samples from all Recipients and Donors were tested in triplicate to assess hemagglutination inhibition (HAI) antibody titers against influenza virus strains included in the 2023–2024 influenza vaccine: A/Victoria/4897/22 (H1N1), A/Darwin/6/21 (H3N2), and B/Austria/1359417/21. An enzyme-linked immunosorbent assay (ELISA) was used to assess the area under the curve (AUC) for binding antibody responses against A/Victoria/4897/22 H1 and A/Darwin/6/21 H3. Further details on assay procedures are provided in in Supporting Information (S4 Text).

## Data analysis

The primary outcome of this study was SAR of influenza virus infections among the Recipients, including virological confirmation, symptomatic infections, or serological evidence of an infection.

Descriptive analyses were conducted to summarize the characteristics of both Donors and Recipients. Violin plots and Loess-smoothed curves were used to visualize viral loads in Donor samples and their temporal trends. Heatmaps and line plots illustrated maximum individual symptom score distributions and changes in total symptom scores over time for both Donors and Recipients. To assess immune responses, HAI titers against vaccine virus strains were plotted to show changes over time. Line plots were also used to depict environmental factors, including indoor temperature, relative humidity, and $CO_2$ concentration. Statistical analyses were done in R and RStudio.

## Supporting information

**S1 Text. EMIT-2 study team.**
(DOCX)

**S2 Text. Acknowledged contributors.**
(DOCX)

**S3 Text. Eligibility criteria.**
(DOCX)

**S4 Text. Supplementary methods.**
(DOCX)

**S1 Table. All Recipients demographics.**
(DOCX)

**S2 Table. All Donors demographics.**
(DOCX)

**S3 Table. Exposure events & activities.**
(DOCX)

**S4 Table. Summary of Donors cough and sneeze counts during 30-minute breath sample sampling.**
(DOCX)

**S1 Fig. Consort diagram for Recipients.** Three Recipients from Cohort 23a re-enrolled in 2024. Two initially joined Cohort 24a, but one was discharged before the study began due to a respiratory infection other than influenza and later re-enrolled in Cohort 24c. Another Recipient from Cohort 23a also joined Cohort 24c. As a result, the total number of distinct individuals who entered quarantine was 27, rather than the summed total (31) from each individual cohort. Note: Influenza Donors were only enrolled in cohorts 24b and 24c; no influenza Donors were enrolled in 23a or 24a.
(DOCX)

**S2 Fig. Consort diagram for Donors.** a. "Phone call received" includes inquiries from social media, flyers, television, and other advertisements. b. Many individuals were ineligible due to lack of influenza infection or were unavailable to commit to the quarantine period. c. Ineligibility among screened individuals included symptom onset >48 hours or underlying medical conditions that did not meet eligibility criteria (S3 Text). d. The Donor enrolled in cohort 24a, recruited from among the phone calls received, tested positive for a seasonal coronavirus (229E); no Donors were recruited for cohort 23a.
(DOCX)

**S3 Fig. Lack of correlation of Donor exhaled breath aerosol viral RNA with viral RNA in either Donor MTS or saliva samples.** Scatterplots show pairwise correlations of viral RNA concentrations measured from exhaled breath aerosol samples (left column: particles <5 μm; right column: particles ≥5 μm) against matched MTS (top row) and saliva (bottom row) samples collected from the same Donors on the same days. Each point represents one Donor–day measurement, with colors and shapes denoting individual Donors. Solid points represent samples from Donors with H3N2 infections, while hollow points represent samples from Donors with H1N1 infections. Black lines indicate linear regression fits with 95% confidence intervals. Pearson correlation coefficients (r) and sample sizes (n) are shown within each plot.
(DOCX)

**S4 Fig. Viral load in ambient bioaerosol and surface swab samples.** (a) RNA copies from ambient bioaerosol samples collected using NIOSH BC-251 bioaerosol sampling devices, stratified by size fraction (>4 μm, 1–4 μm, and <1 μm). (b) RNA copies (left y-axis) and infectious viral load (FFU, right y-axis) from surface swabs collected from marker pens, microphones, and tablets. Each point represents one sample. Only one surface sample, collected from a marker pen, contained virus detected by dPCR; it was also positive by viral culture.
(DOCX)

**S5 Fig. ELISA Area Under the Curve (AUC) over time for Donors and Recipients in Cohort 24b.** ELISA AUC values are shown for Donors (solid lines, filled circles) and Recipients (dot-dashed lines, open circles) at four time points: T0 (Recipient screening), T1 (Admission), T2 (Discharge), and T3 (Follow-up). Virus targets are color-coded: A/Victoria/4897/22 (H1N1, blue) and A/Darwin/6/21 (H3N2, green). Each facet represents an individual study participant.
(DOCX)

**S6 Fig. ELISA Area Under the Curve (AUC) over time for Donors and Recipients in Cohort 24c.** ELISA AUC values are shown for Donors (solid lines, filled circles) and Recipients (dot-dashed lines, open circles) at four time points: T0 (Recipient screening), T1 (Admission), T2 (Discharge), and T3 (Follow-up). Virus targets are color-coded: A/Victoria/4897/22 (H1N1, blue) and A/Darwin/6/21 (H3N2, green). Each facet represents an individual study participant.
(DOCX)

## Acknowledgments

The authors thank the volunteer participants, hotel staff, and all of the research faculty, fellows, and student research assistants of the Public Health AeroBiology Lab, the Center for Sustainability in the Built Environment (City@UMD), the Maryland MEMS & Microfluidics Lab, and Duncan Lab at the University of Maryland, College Park, as well as the clinical staff and research nurses at the University of Maryland, Baltimore, for their contributions to volunteer recruitment, study coordination, sample collection, and laboratory processing (A complete list of contributors can be found in S2 Text). We also thank Bill Lindsley for providing the NIOSH BC-251 bioaerosol samplers used in this project.

## Author contributions

**Conceptualization:** Florian Krammer, Benjamin J. Cowling, Aubree Gordon, Wilbur H. Chen, Jelena Srebric, Donald K. Milton.

**Data curation:** Jianyu Lai, Hamed Sobhani, Filbert Hong.

**Formal analysis:** Jianyu Lai, Hamed Sobhani.

**Funding acquisition:** Don L. DeVoe, Justin R. Ortiz, Shuo Chen, Florian Krammer, Benjamin J. Cowling, Aubree Gordon, Wilbur H. Chen, Jelena Srebric, Donald K. Milton.

**Investigation:** Jianyu Lai, Hamed Sobhani, Kristen K. Coleman, Shih-Han Sheldon Tai, Filbert Hong, Isabel Sierra Maldonado, Yi Esparza, Kathleen M. McPhaul, Shengwei Zhu, Temima Yellin, Juan Manuel Carreno, Wilbur H. Chen.

**Project administration:** Filbert Hong, Isabel Sierra Maldonado, Yi Esparza.

**Software:** Filbert Hong.

**Supervision:** Florian Krammer, Wilbur H. Chen, Jelena Srebric, Donald K. Milton.

**Validation:** Jianyu Lai, Hamed Sobhani, Filbert Hong, Shengwei Zhu, Temima Yellin.

**Visualization:** Jianyu Lai, Hamed Sobhani, Isabel Sierra Maldonado.

**Writing – original draft:** Jianyu Lai, Hamed Sobhani, Kristen K. Coleman, Shih-Han Sheldon Tai, Temima Yellin, Donald K. Milton.

**Writing – review & editing:** Jianyu Lai, Hamed Sobhani, Kristen K. Coleman, Shih-Han Sheldon Tai, Filbert Hong, Isabel Sierra Maldonado, Yi Esparza, Kathleen M. McPhaul, Shengwei Zhu, Don L. DeVoe, Justin R. Ortiz, Shuo Chen, Temima Yellin, Juan Manuel Carreno, Florian Krammer, Benjamin J. Cowling, Aubree Gordon, Wilbur H. Chen, Jelena Srebric, Donald K. Milton.

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
