## [Decision Letter · Decision Letter 0]

27 Jul 2025

Evaluating Modes of Influenza Transmission (EMIT-2): Challenges of a Controlled Human Influenza Virus Infection Transmission Trial (CHIVITT)

PLOS Pathogens

Dear Dr. Milton,

Thank you for submitting your manuscript to PLOS Pathogens. After careful consideration, we feel that it has merit but does not fully meet PLOS Pathogens's publication criteria as it currently stands. Therefore, we invite you to submit a revised version of the manuscript that addresses the points raised during the review process.

Please submit your revised manuscript within 60 days Sep 25 2025 11:59PM. If you will need more time than this to complete your revisions, please reply to this message or contact the journal office at plospathogens@plos.org. Please include the following items when submitting your revised manuscript:

We look forward to receiving your revised manuscript.

Kind regards,

Luis Martínez-Sobrido

Academic Editor

PLOS Pathogens

Matthias Schnell

Section Editor

PLOS Pathogens

Editor-in-Chief

PLOS Pathogens

orcid.org/0000-0003-2946-9497

Editor-in-Chief

PLOS Pathogens

orcid.org/0000-0002-7699-2064

**Additional Editor Comments (if provided):**

**Journal Requirements:**

**Reviewers' Comments:**

Reviewer's Responses to Questions

**Part I - Summary**

Reviewer #1: In the manuscript by Lai et al ‘Evaluating Modes of Influenza Virus Infection Transmission (EMIT-2): Challenges of a Controlled Human Influenza Virus Infection Transmission Trial (CHIVITT)’, the authors set out to establish a human transmission model using donors with community acquired infections. Prior studies by this team and others using experimentally infected donors have resulted in no transmission, and the authors set out to overcome hurdles with virus strain and ventilation by using a poorly ventilated hotel with naturally infected donors. Unfortunately, of the 4 cohorts the team set out to do, only 2 had both donors and recipients enrolled and even then, no human-to-human transmission of influenza virus was observed. While very disappointing, this observation may reveal something more important about community transmission of influenza viruses. However, the paper, in its current form, fails to analyze the results with this lens and is largely a reporting of negative data. My suggestions to the authors emphasize ways in which the data should be restructured and extended to move the field forward and provide larger insight into influenza virus transmission.

A general comment is that the overall pitch of this manuscript is on the original intent of the project (an intervention study) but this is not what the authors were able to do given the lack of transmission. I would suggest that the authors refrain using language that implies that this study accomplishes this goal (see lines 32-33 abstract, author summary, lines 107-109, and largely in the discussion). This will require a considerable rewrite of the manuscript. The cost of these studies is large and therefore it may be hard to separate out the original intent from the observations that can be most useful for a field when the desired outcome is negative.

Reviewer #2: In this study, a controlled human influenza virus infection transmission trial was conducted to study its transmission route and transmission dynamics, which contributed to the development and implementation of prevention and control strategies.

However, several areas require attention to ensure the manuscript meets the standards for publication.

Reviewer #3: The study by Lei et al. represents an interesting attempt to experimentally investigate the relative contributions of different influenza transmission modes, airborne, droplet, and contact under controlled conditions. The authors established a quarantine hotel environment to simulate real-world close-contact exposure between donors with community-acquired influenza and healthy recipients.

A key strength of the study lies in its ambitious design, aiming to experimentally replicate influenza transmission in a naturalistic yet controlled environment. The integration of multiple biological sampling methods across personal, ambient, and surface domains enhances the robustness of the environmental virological data. Furthermore, the authors transparently report challenges, including variable donor shedding, pre-existing recipient immunity, and logistical limitations imposed by influenza season timing and recruitment constraints.

A major limitation is the lack of observed transmission to any recipients. This outcome, while informative, limits the ability to draw firm conclusions about the relative importance of different transmission routes. However, this null result is itself valuable, as it highlights key design barriers that must be addressed in future transmission studies. The study’s findings underscore the importance of early-stage donor recruitment, accurate immunity screening, and increasing cohort size to improve transmission observation power.

Overall, the study will be of interest to researchers and readers, and it provides a critical foundation for refining human challenge transmission models to better understand and mitigate influenza spread.

Reviewer #4: This manuscript by Lai and colleagues from the EMIT-2 Study Team analyzes and discusses the challenges of a controlled human Influenza virus infection transmission trial (CHIVITT). The study design involved donors with community-acquired influenza infections and recipients in the quarantine hotel rooms. The comprehensive approach investigated the transmission modes, including analyzing fine and coarse aerosol, mid-turbinate swabs and saliva using RT-PCR and virus culture. There is no seroconversion occurred (recipient got infected) after the controlled exposure during the trial.

The objective of this study has the potential to provide significant insights into viral transmission at community scale and outside lab environment and should attract broader scientific interest. Compared to the previous EMIT-1 study from the study team, the current study includes donors with community-acquired infections rather than challenge infections. The current design offers enhanced comprehensiveness through monitoring and analysis of both environmental samples and biological samples.

While the limitations of this study are acknowledged in the discussion, they significantly impact the interpretation and significance of the results. The authors have conducted power analysis, but the number of the participants are less than the target sample size. The results indicate no recipient infection. It is then unclear to conclude if this outcome is due to the limited number of recipients or reflects other factors on the virus transmission efficiency. Lack of positive recipient may also undermine the anticipated primary outcome to evaluate modes of the transmission.

Reviewer #5: This study presents a controlled human investigation conducted 2023-2024 influenza season to better understand airborne influenza transmission. The authors recruited five influenza-confirmed patients as donors (3 with H3N2 and 2 with H1N1) and exposed them to 11 individuals under monitored conditions. Clinical, virological, and serological data were collected, demonstrating that all five donors had detectable viral RNA, while four of the five had infectious virus in nasal swabs and actively shed viral RNA. However, only one of these donors shed detectable infectious viral particles. Viral RNA was also detected in air samples. The results showed that none of the exposed individuals became infected.

Overall, this is a well-conducted and clearly presented study that illustrates the complexity of influenza transmission in humans and provides valuable data to the field, despite the small sample size and the absence of transmission events. However, several clarifications are needed:

1) The Methods section should include additional detail. For example, the authors need to clarify whether the digital PCR assay targets the M gene, and whether this refers to digital droplet PCR. They also need to provide more explanation on how fluorescent focus units (FFU) were quantified and whether conventional TCID50 or plaque assays (PFU) were also performed. Citation is need to support FFU can be used to detect the infectious influenza virus particles. They need to clarify how the RNA concentration per cubic meter of air was calculated. The ELISA setup needs to be detailed, including which antigens were used.

2) The detection limits for all of these assays need to be presented, and figures showing both viral RNA and infectious particle data should be included to support interpretation.

3) The authors need to clarify whether the two positive samples originated from the same room and the same individual (lines 202–205).

4) The color schemes used in the figures could be improved for clarity. The current shades of blue and green, as well as red and purple, are difficult to distinguish—especially in the legends. The authors may consider using different shapes (e.g., circles vs. squares) or line styles to enhance visual separation.

**Part II – Major Issues: Key Experiments Required for Acceptance**

Reviewer #1: Suggestions for improvement:

1. Focus on the last 2 cohorts that had successful enrollment of both donors and recipients. The other cohorts can be acknowledged in the MM and consort diagram but it does not need to be implied that data will be presented from 4 cohorts when only 2 are meaningful.

2. It is meaningful that forward transmission was not observed and some ways to expand on that include providing more information about the exposure events and additional analysis/information as the to the types of activities performed, the behaviors between donors and recipients etc. In addition, a few specific comments on the currently presented data are:

a. How many donors and recipients are in each exposure window detailed in Figure 1.

b. Lines 152 and 153 indicate two extra exposure events with only recipients, but the timing of these events relative to the transmission events with donors included is not described in the text or indicated on the timeline in Figure 1. If these exposure events are important enough to warrant inclusion, they should be described in more detail.

c. In exposure conditions – include why these parameters are important to measure in the text and provide information about the various exposure window on the graph or activities to put the values into context for the exposure event.

d. Please note two staff observers when describing the number of occupants in the exposure events in lines 164 and 165. It was not immediately clear why these numbers did not match the number of Donor and Recipient participants.

e. Specify the surfaces swabbed and which one was positive (line 210).

3. The lack of infectious virus in expelled aerosols is interesting and should be further expanded.

a. Explore the data comparing the H3 infected to the H1 infected?

b. Did the expelled virus correlate with saliva or nasal swab titers?

c. Line 135 states there is evidence indicating that Donor D24c-3 had earlier onset, however this evidence is never provided. Please provide evidence for this statement.

4. The difficulty with human transmission studies is surprising given the spread in communities. Please expand upon the inclusion and exclusion criteria for both recipients and donors. Address:

a. During donor screening, there was a low probability that a potential subject would be eligible to participate in the study. There is little discussion of the reasons why most subjects were deemed ineligible. These reasons could be useful to the discussion of why carrying out this type of research is difficult.

b. Were antibody levels or recent vaccination used as exclusion criteria for recipients?

Reviewer #2: The analysis of the results is currently a weak point of the article. The biggest problem with this article is that there is no differential comparison and correlation analysis of the results of several transmission routes. Since it is a study of transmission dynamics and transmission routes, the differences and correlation analyses between samples such as aerosols and surface swabs that highlight different modes of transmission in this study can better reflect which methods of transmission of influenza viruses are predominantly transmitted.

Reviewer #3: 1) The role of pre-existing immunity appears to be important to the lack of transmission, yet the discussion around recipient susceptibility could be expanded. In addition to HAI titres and binding antibody levels, the authors should more thoroughly examine whether mucosal immunity (e.g., sIgA, if available) or T-cell responses might have contributed to protection. I appreciate that new assays can't be added, a deeper discussion of immune correlates and limitations of the current serological data would be valuable.

2) The limited virus shedding by donors is cited as a key challenge. However, the manuscript would benefit from more detailed shedding profiles, including peak shedding window relative to exposure events and whether donors were within the expected infectious period during interactions. A figure aligning symptoms, viral loads and exposure windows would help contextualise the risk of transmission.

3) While viral RNA was detected in aerosols and on surfaces, it would be valuable if the authors could describe whether detected quantities approach infectious doses based on prior literature or other models. Highlighting thresholds of concern would help interpret the biological plausibility of transmission under these conditions, even in the absence of recipient infection.

Reviewer #4: (No Response)

Reviewer #5: N/A

**Part III – Minor Issues: Editorial and Data Presentation Modifications**

Reviewer #1: 1. There are several potential issues with the data presentation in Figures 5 and 6. A). Some of the y-axis legends could be more informative. It is unclear whether the RNA copies are per dPCR reaction, per sample, per volume unit, or some other unit. Likewise, it is unclear what the denominators for the FFU values are. B). It is unclear why the day values on the x-axis are whole numbers in some panels and non-whole numbers in the Loess plots in panel B. Additionally, it is unclear what the purpose of the Loess smoothed curves is. C). Consider making the y-axis scales consistent in panels c, d, e, and f to make it easier to compare the levels in the different sample types.

2. Key details regarding the dPCR assay, including the gene target and sequences, should be included or have a prior publication referenced.

3. Consort diagrams in Fig 2 and 3 can be combined to demonstrate that cohorts 23a and 24a failed to enroll donors.

Reviewer #2: Line 91-93 The author should show the extensive research in introduction.

Result: In line 113, in the time graph of the queue, the author is advised to modify the image to add the corresponding time unit.

This modification will help improve the intuitiveness of the graphical representation。

Study cohort and study population: In line 125, the concept of IR appears for the first time, and I suggest that the author explain the role of IR when the concept of IR first appears, rather than just in the conclusion part, to increase the comprehensibility and smoothness of the article.

Exhaled Breath Aerosol Samples: On line 178, 8.9x103 copies should be changed to 8.9×103 copies.

The symbol "×" is used to denote the multiplication of numbers, which is different from the Roman numeral "x".

There are several places in the article about this symbol, and I recommend that the author review and standardize the unit symbol throughout the manuscript to maintain consistency in the use of units.

Surface Swab Viral Load: In this subsection, it is recommended that the author add a graph on surface swab load. This provides a more visual indication of the propagation strength of the mode of propagation represented by the surface swab.

It is suggested that the author add a diagram illustrating the overall experimental results and a model for future epidemic prevention and control.

Reviewer #3: Line 39: “florescent focus assays”, should be fluorescent focus assay

Line 40: add abbreviation (ELISA)

Line 44: there is no description what NIOSH is.

Line 351-352: “CRs did not wear face shield or sanitize their hands during the exposure event; they would have been exposed to all modes of transmission”. Was the behaviour of the CRs influenced by their awareness of the infection risk? For example, did they consciously avoid touching their face, compared to typical everyday behaviour in individuals who are not actively considering infection risk? Given the high frequency with which people typically touch their faces per hour, this could be an important factor affecting the likelihood of contact transmission.

Line 394: “score throat” should be sore throat

Figure 5 caption: a) put abbreviation for mid-turbinate swabs (MTS), “aaliva” should be saliva

Reviewer #4: 1. The authors have mentioned that compared to EMIT-1, EMIT-2 recruited symptomatic community-acquired donor to achieve expected SAR. However, the current study lacking statistically significant data, could not provide a direct comparison or evaluation of this under this experimental setup. In the future study, increasing the number of donors to reach the expected power is necessary. Given the inherent limitations in recruiting sufficient community-acquired donors within specific timeframes and given periods that possibly missing the peak viral loads, will the authors consider alternative ways like those in EMIT-1 to increase the number?

2. The authors mentioned that lower overall contagiousness such as less cough limited the current results. However, again due to the insufficient statistical power, this could not be statistically tested from the current study. Is it feasible or ethically acceptable to establish and request a minimal number of coughs for certain donors during the exposure in comparison to the exposure where cough barely occurred.

3. Figure 1 and 4 require detailed captions. It would be helpful and clear to indicate the exposure event periods in Figure 4 and label the recipients who have received the vaccinations in Figure 9 and 10. Typo in Figure 5: saliva

Reviewer #5: 1) The Methods section should include additional detail. For example, the authors need to clarify whether the digital PCR assay targets the M gene, and whether this refers to digital droplet PCR. They also need to provide more explanation on how fluorescent focus units (FFU) were quantified and whether conventional TCID50 or plaque assays (PFU) were also performed. Citation is need to support FFU can be used to detect the infectious influenza virus particles. They need to clarify how the RNA concentration per cubic meter of air was calculated. The ELISA setup needs to be detailed, including which antigens were used.

2) The detection limits for all of these assays need to be presented, and figures showing both viral RNA and infectious particle data should be included to support interpretation.

3) The authors need to clarify whether the two positive samples originated from the same room and the same individual (lines 202–205).

4) The color schemes used in the figures could be improved for clarity. The current shades of blue and green, as well as red and purple, are difficult to distinguish—especially in the legends. The authors may consider using different shapes (e.g., circles vs. squares) or line styles to enhance visual separation.

PLOS authors have the option to publish the peer review history of their article (what does this mean? ). If published, this will include your full peer review and any attached files.

**Do you want your identity to be public for this peer review?** For information about this choice, including consent withdrawal, please see our Privacy Policy .

Reviewer #1: No

Reviewer #2: No

Reviewer #3: No

Reviewer #4: No

Reviewer #5: No

**Figure resubmission:**

**Reproducibility:**



---

## [Decision Letter · Decision Letter 1]

16 Nov 2025

PPATHOGENS-D-25-00991R1

Evaluating Modes of Influenza Transmission (EMIT-2): Insights from a Controlled Human Influenza Virus Infection Transmission Trial (CHIVITT)

PLOS Pathogens

Dear Dr. Milton,

Thank you for submitting your manuscript to PLOS Pathogens. After careful consideration, we feel that it has merit but does not fully meet PLOS Pathogens's publication criteria as it currently stands. Therefore, we invite you to submit a revised version of the manuscript that addresses the points raised during the review process.

We look forward to receiving your revised manuscript.

Kind regards,

Luis Martínez-Sobrido

Academic Editor

PLOS Pathogens

Matthias Schnell

Section Editor

PLOS Pathogens

Sumita Bhaduri-McIntosh

Editor-in-Chief

PLOS Pathogens

orcid.org/0000-0003-2946-9497

Michael Malim

Editor-in-Chief

PLOS Pathogens

orcid.org/0000-0002-7699-2064

**Journal Requirements:**

**Reviewers' Comments:**

Reviewer's Responses to Questions

**Part I - Summary**

Reviewer #1: In the revised manuscript by Lai et al ‘Evaluating Modes of Influenza Virus Infection Transmission (EMIT-2): Challenges of a Controlled Human Influenza Virus Infection Transmission Trial (CHIVITT)’, the authors set out to establish a human transmission model using donors with community acquired infections. They restructuring and describing of the negative data is more impactful and raises interesting questions about community transmission.

Reviewer #3: All corrections requested were addressed to my satisfaction.

Reviewer #4: In this revised manuscript, the authors have undertaken extensive revisions to the results and discussion sections. This is actually a substantial experimental project involving many factors in the experimental design. Therefore, the detailed information provided in the revision offers insightful perspectives and valuable experience that will benefit future experimental design in this field.

The authors have also addressed my previous questions well in their revision.

Reviewer #5: The manuscript has been significantly improved.

**Part II – Major Issues: Key Experiments Required for Acceptance**

Reviewer #1: (No Response)

Reviewer #3: (No Response)

Reviewer #4: N/A

Reviewer #5: (No Response)

**Part III – Minor Issues: Editorial and Data Presentation Modifications**

Reviewer #1: A few suggestions remain, mostly for increased readability and topics to cover in the discussion.

1. The title does not accurately describe the results, consider revising.

2. Figure inclusion and structure:

a. The consort diagrams (Fig 2 and 3) are not very meaningful (expect that recruitment from campus testing kiosk was imperative for identifying eligible donors). Consider moving to the supplement.

b. The S1, S2, S3, and S4 text are confusing. S1 text includes many of the authors – why is this a separate list? Also it is not referenced anywhere that I could tell. S2 text should be in the acknowledgments and S3 and S4 text should be combined into one document.

3. Correlation graphs in figure S1:

a. The regressions with the aerosol >5 micron are driven by a single positive point and thus not accurate.

b. Please add more descriptive labels to the axis.

4. The three reasons given for the lack of transmission stimulate the following topics that could be expanded in the discussion:

a. Typical indoor environmental conditions (temp, humidity, and ACH) where transmission is known to occur.

b. The heterogeneity in expulsion of viruses into the air. This is mentioned a bit, but beyond the work done by this team, SARS-CoV-2 controlled infections have revealed a heterogeneity in expulsion of virus-containing aerosols.

Reviewer #3: (No Response)

Reviewer #4: N/A

Reviewer #5: (No Response)

PLOS authors have the option to publish the peer review history of their article (what does this mean? ). If published, this will include your full peer review and any attached files.

**Do you want your identity to be public for this peer review?** For information about this choice, including consent withdrawal, please see our Privacy Policy .

Reviewer #1: No

Reviewer #3: No

Reviewer #4: No

Reviewer #5: No

**Figure resubmission:**
---

## [Decision Letter · Decision Letter 2]

24 Dec 2025

Dear Dr. Milton,

We are pleased to inform you that your manuscript 'Evaluating Modes of Influenza Transmission (EMIT-2): Insights from a Controlled Human Influenza Virus Infection Transmission Trial (CHIVITT)' has been provisionally accepted for publication in PLOS Pathogens.

Best regards,

Luis Martínez-Sobrido

Academic Editor

PLOS Pathogens

Matthias Schnell

Section Editor

PLOS Pathogens

Sumita Bhaduri-McIntosh

Editor-in-Chief

PLOS Pathogens

orcid.org/0000-0003-2946-9497

Michael Malim

Editor-in-Chief

PLOS Pathogens

orcid.org/0000-0002-7699-2064

Reviewer Comments (if any, and for reference):

Reviewer's Responses to Questions

**Part I - Summary**

Reviewer #1: The authors have address all the substantive concerns - but to this reviewer the title is still not accurate, perhaps because of the ues of the EMIT acronym 'Evaluating Modes of Influenza Transmission (EMIT-2): Insights from a Controlled Human Influenza Virus Infection Transmission Trial (CHIVITT)' - The work does not evaluate the modes of transmission because there was not any transmission, and it is not a controlled human infection model thus it is a bit misleading. A more accurate title would be: "Lack of Transmission from Naturally Acquired Seasonal Influenza Infections in a Controlled Setting" or a title based on the conclusions from characterization of the naturally infected donors.

**Part II – Major Issues: Key Experiments Required for Acceptance**

Reviewer #1: (No Response)

**Part III – Minor Issues: Editorial and Data Presentation Modifications**

Reviewer #1: (No Response)

PLOS authors have the option to publish the peer review history of their article (what does this mean? ). If published, this will include your full peer review and any attached files.

**Do you want your identity to be public for this peer review?** For information about this choice, including consent withdrawal, please see our Privacy Policy .

Reviewer #1: No

---

## [Editor Report · Acceptance letter]

Dear Dr. Milton,

We are delighted to inform you that your manuscript, "Evaluating Modes of Influenza Transmission (EMIT-2): Insights from Lack of Transmission in a Controlled Transmission Trial with Naturally Infected Donors," has been formally accepted for publication in PLOS Pathogens.

Best regards,

Sumita Bhaduri-McIntosh

Editor-in-Chief

PLOS Pathogens

orcid.org/0000-0003-2946-9497

Michael Malim

Editor-in-Chief

PLOS Pathogens

orcid.org/0000-0002-7699-2064